# What Makes Objects Similar:
# A Unified Multi-Metric Learning Approach

**Han-Jia Ye    De-Chuan Zhan    Xue-Min Si    Yuan Jiang    Zhi-Hua Zhou**
National Key Laboratory for Novel Software Technology,
Nanjing University, Nanjing, 210023, China
{yehj,zhandc,sixm,jiangy,zhouzh}@lamda.nju.edu.cn

## Abstract

Linkages are essentially determined by similarity measures that may be derived from multiple perspectives. For example, spatial linkages are usually generated based on localities of heterogeneous data, whereas semantic linkages can come from various properties, such as different physical meanings behind social relations. Many existing metric learning models focus on spatial linkages, but leave the *rich* semantic factors unconsidered. Similarities based on these models are usually overdetermined on linkages. We propose a Unified Multi-Metric Learning ($\text{UM}^2\text{L}$) framework to exploit multiple types of metrics. In $\text{UM}^2\text{L}$, a type of combination operator is introduced for distance characterization from multiple perspectives, and thus can introduce flexibilities for representing and utilizing both spatial and semantic linkages. Besides, we propose a uniform solver for $\text{UM}^2\text{L}$ which is guaranteed to converge. Extensive experiments on diverse applications exhibit the superior classification performance and comprehensibility of $\text{UM}^2\text{L}$. Visualization results also validate its ability on physical meanings discovery.

## 1  Introduction

Similarities measure the closeness of connections between objects and usually are reflected by distances. Distance Metric Learning (DML) aims to learn appropriate metric that can figure out the underlying linkages or connections, thus can greatly improve the performance of similarity-based classifiers, such as $k$NN.

Objects are linked with each other for different reasons. Global DML methods consider the deterministic single metric which measures similarities between all object pairs. Recently, investigations on local DML have considered locality specific approaches, and consequently multiple metrics are learned. These metrics are either in charge of different spatial areas [15, 20] or responsible for each specific instance [7, 22]. Both global and local DML methods emphasize the linkage constraints (including must-link and cannot-link) in localities with univocal semantic meaning, e.g., the side information of class. However, there can be many different reasons for two instances to be similar in real world applications [3, 9].

Linkages between objects can be with multiple latent semantics. For example, in a social network, friendship linkages may lie on different hobbies of users. Although a user has many friends, their common hobbies could be different and as a consequence, one can be friends with others for different reasons. Another concrete example is, for articles on "A. Feature Learning" which are closely related to both "B. Feature Selection" and "C. Subspace Models", their connections are different in semantics. The linkage between A and B emphasizes "picking up some helpful features", while the common semantic between A and C is about "extracting subspaces" or " feature transformation". These phenomena clearly indicate ambiguities rather than a single meaning in linkage generation.

Hence, the distance/similarity measurements are overdetermined in these applications. As a consequence, a new type of multi-metric learner which can describe the ambiguous linkages is desired.

In this paper, we propose a Unified Multi-Metric Learning ($\text{UM}^2\text{L}$) approach which integrates the consideration of linking semantic ambiguities and localities in one framework. In the training process, more than one metric is learned to measure distances between instances and each of them reflects a type of inherent spatial or semantic properties of objects. During the test, $\text{UM}^2\text{L}$ can automatically pick up or integrate these measurements, since semantically/spatially similar data points have small distances and otherwise they are pulled away from each other; such a mechanism enables the adaptation to environment to some degree, which is important for the development of learnwares [25]. Furthermore, the proposed framework can be easily adapted to different types of ambiguous circumstances: by specifying the mechanism of metric integration, various types of linkages in applications can be considered; by incorporating sparse constraints, $\text{UM}^2\text{L}$ also turns out good visualization results reflecting physical meanings of latent linkages between objects; besides, by limiting the number of metrics or specifying the regularizer, the approach can be degenerated to some popular DML methods, such as MMLMNN [20]. Benefitting from alternative strategy and stochastic techniques, the general framework can be optimized steadily and efficiently.

Our main contributions are: **(I)** A Unified Multi-Metric Learning framework considering both data localities and ambiguous semantics linkages. **(II)** A flexible framework adaptable for different tasks. **(III)** Unified and efficient optimization solutions, superior and interpretable results.

The rest of this paper starts with some notations. Then the $\text{UM}^2\text{L}$ framework is presented in detail, which is followed by a review of related work. The last are experiments and conclusion.

## 2 The Unified Multi-Metric Framework

Generally speaking, the supervision information for Distance Metric Learning (DML) is formed as pairwise constraints or triplet sets. We restrict our discussion on the latter one, $\mathcal{T} = \{\mathbf{x}^t, \mathbf{y}^t, \mathbf{z}^t\}_{t=1}^T$, since it provides more local information. In each triplet, target instance $\mathbf{y}^t$ is more similar to $\mathbf{x}^t$ than imposter $\mathbf{z}^t$ and $\{\mathbf{x}^t, \mathbf{y}^t, \mathbf{z}^t\} \in \mathbb{R}^d$. $\mathcal{S}_d$ and $\mathcal{S}_d^+$ are the set of symmetric and positive semi-definite (PSD) matrix of size $d \times d$, respectively. $I$ is the identity matrix. Matrix Frobenius Norm $\|M\|_F = \sqrt{\text{Tr}(M^\top M)}$. Let $\boldsymbol{m}_i$ and $\boldsymbol{m}^j$ denote the $i$-th row and $j$-th column of matrix $M$ respectively, and $\ell_{2,1}$-norm $\|M\|_{2,1} = \sum_i^d \|\boldsymbol{m}_i\|_2$. Operator $[\cdot]_+ = \max(\cdot, 0)$ preserves the non-negative part of the input value. DML aims at learning a metric $M \in \mathcal{S}_d^+$ making similar instances have small distances to each other and dissimilar ones far apart. The (squared) Mahalanobis distance between pair $(\mathbf{x}^t, \mathbf{y}^t)$ with metric $M$ can be denoted as:

$$\text{Dis}_M^2(\mathbf{x}^t, \mathbf{y}^t) = (\mathbf{x}^t - \mathbf{y}^t)^\top M(\mathbf{x}^t - \mathbf{y}^t) = \text{Tr}(MA_{\mathbf{xy}}^t). \tag{1}$$

$A_{\mathbf{xy}}^t = (\mathbf{x}^t - \mathbf{y}^t)(\mathbf{x}^t - \mathbf{y}^t)^\top \in \mathcal{S}_d^+$ is the outer product of difference between instance $\mathbf{x}^t$ and $\mathbf{y}^t$. The distance in Eq.1 assumes that there is a *single* type of relationship between object features, which uses univocal linkages between objects.

Multi-metric learning takes data heterogeneities into consideration. However, both single metric learned by global DML and multiple metrics learned with local methods focus on exploiting locality information, i.e., constraints or metrics are closely related to the localities. In particular, local DML approaches mainly aim at learning a set of multiple metrics one for each local area. In this paper, a general multi-metric configuration is investigated to deal with linkage ambiguities from both semantic and locality perspectives. We denote the set of $K$ multiple metrics to be learned as $\mathcal{M}_K = \{M_1, M_2, \ldots, M_K\}$ and $\{M_k\}_{k=1}^K \in \mathcal{S}_d^+$. Similarity score between a pair of instances based on $M_k$, w.l.o.g., can be set as the negative distance, i.e., $f_{M_k}(\mathbf{x}^t, \mathbf{y}^t) = -\text{Dis}_{M_k}^2(\mathbf{x}^t, \mathbf{y}^t)$. In multi-metric scenario, consequently, there will be a set of multiple similarity scores $f_{\mathcal{M}_K} = \{f_{M_k}\}_{k=1}^K$. Each metric/score in the set reflects a particular semantic or spatial view of data. The *overall* similarity score $f^v(\mathbf{x}^t, \mathbf{y}^t) = \kappa_v(f_{\mathcal{M}_K}(\mathbf{x}^t, \mathbf{y}^t)), v = \{1, 2\}$ and $\kappa_v(\cdot)$ is a *functional operator* closely related to concrete applications, which maps the set of similarity scores w.r.t. all metrics to a single value.

With these discussions, the Unified Multi-Metric Learning ($\text{UM}^2\text{L}$) framework can be denoted as:

$$\min_{\mathcal{M}_K} \frac{1}{T} \sum_{t=1}^T \ell\left(f^1(\mathbf{x}^t, \mathbf{y}^t) - f^2(\mathbf{x}^t, \mathbf{z}^t)\right) + \lambda \sum_{k=1}^K \Omega_k(M_k). \tag{2}$$

The overall inter-instance similarity $f^1$ and $f^2$ are based on operators $\kappa_1$ and $\kappa_2$ respectively. $\ell(\cdot)$ is a convex loss function which encourages $(\mathbf{x}^t, \mathbf{y}^t)$ to have larger overall similarity score than $(\mathbf{x}^t, \mathbf{z}^t)$. Note that although inter-instance similarities are defined on different metrics in $\mathcal{M}_K$, the convex loss function $\ell(\cdot)$ acts as a bridge and makes the similarities measured by different metrics comparable as in [20]. The fact that triplet restrictions being provided without specifying concrete measurements makes it reasonable to use flexible $\kappa$s. For instance, in a social network, similar nodes only share some common interests (features) rather than consistently possessing all interests. Tendency on different types of hobbies can be reflected by various metrics. Therefore, the similarity scores may be calculated with different measurements and operator $\kappa_v$ is used for taking charge of "selecting" or "integrating" the right base metric for measuring similarities. The choices of loss functions and $\kappa$s are substantial issues in this framework and will be described later. Convex regularizer $\Omega_k(M_k)$ can impose prior or structure information on base metric $M_k$. $\lambda \geq 0$ is a balance parameter.

## 2.1 Choices for $\kappa$

$\textsc{Um}^2\textsc{l}$ takes both *spatial* and ambiguous *semantic* linkages into account based on the configurations of $\kappa$, which integrates or selects base metrics. As an integrator, in applications where locality related multiple metrics are needed, $\kappa$ can be an RBF like function which decreases as the distance is increasing. The locality determines the impact of each metric. When $\kappa$ acts as a selector, $\textsc{Um}^2\textsc{l}$ should *automatically* assign triplets to one of the metrics which can explain instance similarity/dissimilarity best. Besides, from the aspect of loss function $\ell(\cdot)$, the elected $f$s form a comparable set of similarity measurements [17, 20]. In this case, we may implement the operator $\kappa$ by choosing the most remarkable base metric making the pair of instances $\mathbf{x}^t$ and $\mathbf{y}^t$ similar. Advantages of selection mechanism are two folds. First, it reduces the impact of initial triplets construction in localities [19]; second, it stresses the most evident semantic and reflects the consideration of ambiguous semantics in a linkage construction. Choices of $\kappa$s heavily depend on concrete applications. It is actually a combiner and can get inspiration from ensemble methods [24]. Here, we mainly consider 4 different types of linkage based on various sets of $\kappa$s as follows.

**Apical Dominance Similarity (ADS)**: which is named after the phenomenon in auxanology of plants, where the most important term dominates the evaluation. In this case, $\kappa_1 = \kappa_2 = \max(\cdot)$, i.e., maximum similarity among all similarities calculated with $\mathcal{M}_K$ on similar pair $(\mathbf{x}^t, \mathbf{y}^t)$ should be larger than the maximum similarity of $(\mathbf{x}^t, \mathbf{z}^t)$. This corresponds to similar pairs being close to each other under *at least one* measurement, meanwhile dissimilar pairs are disconnected by *all* different measurements. This type of linkage generation often occurs in social network applications, e.g., nodes are linked together for a portion of similar orientations while nodes are unlinked because there are no common interests. By explicitly modeling each node in a social network as an instance, each of the base metrics $\{M_k\}_{k=1}^K$ can represent parts of semantics in linkages. Then the dissimilar pair in a triplet, e.g., the non-friendship relationship, should be with small similarity scores over $\mathcal{M}_K$; while for the similar pair, there should be at least one base similarity score with high value, which reflects their common interests [3, 11].

**One Vote Similarity (OVS)**: which indicates the existence of potential key metric in $\mathcal{M}_K$, i.e., either similar or dissimilar pair is judged by *at least one* key metric respectively, while remaining metrics with other semantic meanings are ignored. In this case, $\kappa_1 = \max(\cdot)$ and $\kappa_2 = \min(\cdot)$. This type of similarity should usually be applied as an "interpreter" in domains like image, video which are with complicated semantics. The learned metrics reveal different latent concepts in objects. Note that simply applying OVS in $\textsc{Um}_2\textsc{l}$ with impropriate regularizer $\Omega$ will lead to a trivial solution, i.e., $M_k = 0$, which satisfies all similar pair restrictions yet has no generalization ability. Therefore, we need to set $\Omega_k(M_k) = \|M_k - I\|_F^2$ or restrict the trace of $M_k$ to equal to 1.

**Rank Grouping Similarity (RGS)**: which groups the pairs and makes the similar pairs with higher ranks than dissimilar ones. This is the most rigorous similarity and we also refer it as One-Vote Veto Similarity (OV$^2$S). In this case, $\kappa_1 = \min(\cdot)$ while $\kappa_2 = \max(\cdot)$, which regards the pairs as dissimilar even when there is only one metric denying the linkage. This case is usually applied to applications where latent multiple views exist and different views are measured by different metrics in $\mathcal{M}_K$. In these applications, it is obviously required that all potential views obtain consistencies, and weak conflict detected by one metric should also be punished by RGS (OV$^2$S) loss.

**Average Case Similarity (ACS)**: which treats all metrics in $\mathcal{M}_K$ equally, i.e., $\kappa_1 = \kappa_2 = \sum(\cdot)$. This is the general case when there is no prior knowledge on applications.

There are many derivatives of similarity where $\kappa_v$ is configured as $\min(\cdot)$, $\max(\cdot)$ and $\sum(\cdot)$. Furthermore, $\kappa_v$ in fact can be with richer forms, and we will leave the discussions of choosing different $\kappa$s later in section 3. Besides, in the framework, multiple choices of the regularizer $\Omega_k(\cdot)$ can be made. As most DML methods [14], $\Omega_k(M_k)$ can be set as $\|M_k\|_F^2$. Yet it also can be incorporated with more structural information, e.g., we can configure $\Omega(M_k) = \|M_k\|_{2,1}$, where the row/column sparsity filters influential features for composing linkages in a network; or $\Omega_k(M_k) = \text{Tr}(M_k)$, which guarantees the low rank property for all metrics. Due to the high applicability of the proposed framework, we name it as UM$^2$L (Unified Multi-Metric Learning).

## 2.2 General Solutions for UM$^2$L

UM$^2$L can be solved alternatively between metrics $\mathcal{M}_K$ and affiliation portion of each instance, when $\kappa$ is a piecewise linear operator such as $\max(\cdot)$ and $\min(\cdot)$. For example, in the case of ADS, the metric used to measure the similarity of pair $(\mathbf{x}^t, \mathbf{y}^t)$ is decided by: $k_{v,*}^t = \arg\max_k f_{M_k}(\mathbf{x}^t, \mathbf{y}^t)$, which is the index of the metric $M_k$ that has the largest similarity value over the pair. Once the dominating key metric of each instance is found, the whole optimization problem is convex w.r.t. each $M_k$, which can be easily optimized. On account of the convexity of each sub-problem in the alternating approach, the whole objective is ensured to decrease in iterations so as to converge eventually. It is notable that when dealing with a single triplet in a stochastic approach, the convergence can be guaranteed as well in Theorem 1, which will be introduced later.

In batch case, for facilitating the discussion, we can implement $\ell(\cdot)$ as the smooth hinge loss, i.e., $\ell(x) = [\frac{1}{2} - x]_+$ if $x \geq 1$ or $x \leq 0$ and equals to $\frac{1}{2}(1-x)^2$ otherwise. If trace norm $\Omega_k(M_k) = \text{Tr}(M_k)$ is used, $\mathcal{M}_K$ can be solved with accelerated projected gradient descent method. If the whole objective in Eq. 2 is denoted as $\mathcal{L}_{\mathcal{M}_K}$, the gradient w.r.t. one metric $M_k$ can be computed as:

$$\frac{\partial \mathcal{L}_{\mathcal{M}_K}}{\partial M_k} = \frac{1}{T} \sum_{t \in \hat{\mathcal{T}}_k} \frac{\partial \ell(\text{Tr}(M_{k_{2,*}^t} A_{\mathbf{xz}}^t) - \text{Tr}(M_{k_{1,*}^t} A_{\mathbf{xy}}^t))}{\partial M_k} + \lambda I = \frac{1}{T} \sum_{t \in \hat{\mathcal{T}}_k} \nabla_{M_k}^t(a^t) + \lambda I , \qquad (3)$$

where the first part is a sum of gradients over the triplets subset $\hat{\mathcal{T}}_k$ whose membership indexes containing $k$, i.e., $\hat{\mathcal{T}}_k = \{t \mid k = k_{1,*}^t \text{ or } k = k_{2,*}^t\}$. The separated gradient $\nabla_{M_k}^t(a^t)$, with $a^t = \text{Tr}(M_{k_{2,*}^t} A_{\mathbf{xz}}^t) - \text{Tr}(M_{k_{1,*}^t} A_{\mathbf{xy}}^t)$, for triplet $t \in \hat{\mathcal{T}}_k$ is:

$$\nabla_{M_k}^t(a^t) = \begin{cases} 0 & \text{if } a^t \geq 1 \\ \delta(k = k_{1,*}^t) A_{\mathbf{xy}}^t - \delta(k = k_{2,*}^t) A_{\mathbf{xz}}^t & \text{if } a^t \leq 0 \\ \delta(k = k_{1,*}^t)(1 - a^t) A_{\mathbf{xy}}^t - \delta(k = k_{2,*}^t)(1 - a^t) A_{\mathbf{xz}}^t & \text{otherwise} \end{cases} .$$

$\delta(\cdot)$ is the Kronecker delta function, which contributes to the computation of the gradient when $\kappa_v$ is optimized by $M_k$. After accelerated gradient descent, a projection step is conducted to maintain the PSD property of each solution. If structured sparsity is stressed, $\ell_{2,1}$-norm is used as a regularizer, i.e., $\Omega_k(M_k) = \|M_k\|_{2,1}$. FISTA [2] can be used to optimize the non-smooth regularizer efficiently: after a gradient descent with step size $\gamma$ on the smooth loss to get an intermediate solution $V_k = M_k - \gamma \frac{1}{T} \sum_{t \in \hat{\mathcal{T}}_k} \nabla_{M_k}^t(a^t)$, the following proximal sub-problem is conducted to get a further update:

$$M_k' = \arg\min_{M \in \mathcal{S}_d} \frac{1}{2} \|M - V_k\|_F^2 + \lambda \|M\|_{2,1} . \qquad (4)$$

The PSD property of $M_k$ can be ensured by a projection in each iteration, or can often be preserved by last step projection [14]. Hence, in Eq. 4, only symmetric constraint of $M_k$ is imposed. Since $\ell_{2,1}$-norm considers only one-side (row-wise) property of a matrix, Lim *et al.* [12] uses iterative symmetric projection to get a solution, which has heavy computational cost in some cases. In a reweighted way, the proximal subproblem can be tackled by the following lemma efficiently.

**Lemma 1** *The proximal problem in Eq. 4 can be solved by updating diagonal matrixes $D_1$ and $D_2$ and symmetric matrix $M$ alternatively:*

$$\{D_{1,ii} = \frac{1}{2\|\boldsymbol{m}_i\|_2}, \ D_{2,ii} = \frac{1}{2\|\boldsymbol{m}^i\|_2}\}_{i=1}^d \ ; vec(M) = (I \otimes (I + \frac{\lambda}{2}D_1) + (\frac{\lambda}{2}D_2 \otimes I))^{-1} vec(V_k) ,$$

*where $vec(\cdot)$ is the vector form of a matrix and $\otimes$ means the Kronecker product. Due to the* diagonal *property of each term, the update of $M$ can be further simplified.*[1]

The update of $M$ in Lemma 1 takes row-wise and column-wise $\ell_2$-norm into consideration simultaneously, and usually gets converged in about $5 \sim 10$ iterations.

The batch solution for UM²L can benefit from the acceleration strategy [2]. The computational cost of a full gradient, however, sometimes becomes the dominant expense owing to the huge number of triplets. Inspired by [6], we propose a stochastic solution, which manipulates one triplet in each iteration. In the $s$-th iteration, we sample a triplet $(\mathbf{x}^s, \mathbf{y}^s, \mathbf{z}^s)$ uniformly and update current solution set $\mathcal{M}_K^s = \{M_k^s\}_{k=1}^K$. The whole objective of $s$-th iteration with $\mathcal{M}_K^s$ is:

$$\mathcal{L}_{\mathcal{M}_K^s}^s = \ell(f^1(\mathbf{x}^s, \mathbf{y}^s) - f^2(\mathbf{x}^s, \mathbf{z}^s)) + \lambda \sum_{k=1}^K \Omega_k(M_k^s). \tag{5}$$

Similar to proximal gradient solution, after doing (sub-) gradient descent on the loss function in Eq. 5, proximal operator can be utilized to update base metrics $\{M_k^s\}_{k=1}^K$. The stochastic strategy is guaranteed to converge theoretically. By denoting $\mathcal{M}_K^* = (M_1^*, \ldots, M_K^*) \in \arg\min \sum_{s=1}^S \mathcal{L}^s(M_1^s, \ldots, M_K^s)$ as the optimal solution, given totally $S$ iterations, we have:

**Theorem 1** *Suppose in* UM²L *framework, the loss $\ell(\cdot)$ is a convex one and selection operator $\kappa_v$ is in piecewise linear form. If each training instance $\|\mathbf{x}\|_2 \leq 1$, the sub-gradient set of $\Omega_k(\cdot)$ is bounded by $R$, i.e., $\|\partial\Omega_k(M_k)\|_F^2 \leq R^2$ and sub-gradient of loss $\ell(\cdot)$ is bounded by $C$. When for each base metric[2] $\|M_k - M_k^*\|_F \leq D$, it holds that:[3]*

$$\sum_{s=1}^S \mathcal{L}_{\mathcal{M}_K^s}^s - \mathcal{L}_{\mathcal{M}_K^*}^s \leq 2GD + B\sqrt{S}$$

*with $G^2 = \max(C^2, R^2)$ and $B = (\frac{D^2}{2} + 8G^2)$. Given hinge loss, $C^2 = 16$.*

## 3  Related Work and Discussions

Global DML approaches devote to finding a single metric for all instances [5, 20] while local DML approaches further take spatial data heterogeneities into consideration. Recently, different types of local metric approaches are proposed, either *assigning* cluster-specific metric to instance based on locality [20] or *constructing* local metrics generatively [13] or discriminatively [15, 18]. Furthermore, instance specific metric learning methods [7, 22] extend the locality properties of linkages to extreme and gain improved classification performance. However, these DML methods, either global or local, take univocal semantic from *label*, namely, the side information.

Richness of semantics is noticed and exploited by machine learning researchers [3, 11]. In DML community, PSD [9] and SCA [4] are proposed. PSD works as collective classification which is less related to UM²L. SCA, a multi-metric learning method based on pairwise constraints, focuses on learning metrics under one specific type of ambiguities, i.e., linkages are with competitive semantic meanings. UM²L is a more general multi-metric learning framework which considers triplet constraints and *various kinds of* ambiguous linkages from both localities and semantic views.

UM²L maintains good compatibilities and can degenerate to several state-of-the-art DML methods. For example, by considering univocal semantic ($K = 1$), we can get a global metric learning model used in [14]. If we further choose the hinge loss and set the regularizer $\Omega(M) = \text{tr}(MB)$ with $B$ an intra-class similar pair covariance matrix, UM²L degrades to LMNN [20]. With trace norm on $M$, [10] is recovered. For multi-metric approaches, if we set $\kappa_v$ as the indicator of classes for the second instance in a similar or dissimilar pair, UM²L can be transformed to MMLMNN [20].

## 4  Experiments on Different Types of Applications

Due to different choices of $\kappa$s in UM²L, we test the framework in diverse real applications, namely social linkages/feature pattern discovering, classification, physical semantic meaning distinguishing and visualization on multi-view semantic detection. To simplify the discussion, we use alternative batch solver, smooth hinge loss and set regularizer $\Omega_k(M_k) = \|M_k\|_{2,1}$ if without further statement. Triplets are constructed with 3 targets and 10 impostors with Euclidean nearest neighbors.

## 4.1 Comparisons on Social Linkage/Feature Pattern Discovering

ADS configuration is designed for social linkage and pattern discovering. To validate the effectiveness of $\text{UM}^2\text{L}_{\text{ADS}}$, we test it on social network data and synthetic data to show its grouping ability on linkages and features, respectively.

Social linkages come from 6 real world Facebook network datasets from [11]. Given friendship circles of an ego user and users' binary features, the goal of ego-user linkages discovering is to utilize the overall linkage and figure out how users are grouped. We form instances by taking absolute value of differences between features of ego and the others. After circles with $< 5$ nodes are removed, $K$ is configured as the number of circles remained. Pairwise distance is computed by each metric in $\mathcal{M}_K$, and a threshold is tuned on the training set to filter out irrelevant users. Thus, users with different common hobbies are grouped together. MAC detects group assignments based on binary features [8]; SCA constructs user linkages in a probabilistic way, and EGO [11] can directly output user circles. KMeans (KM) and Spectral Clustering (SC) directly group users based on their features without using linkages. Performance is measured by Balanced Error Rate (BER) [11], the lower the better. Results are listed in Table 1, which shows $\text{UM}^2\text{L}_{\text{ADS}}$ performs the best on most datasets.

**Table 1:** BER of the linkage discovering comparisons on Facebook datasets: $\text{UM}^2\text{L}_{\text{ADS}}$ vs. others

| BER↓ | KM | SP | MAC | SCA | EGO | UM²L |
|---|---|---|---|---|---|---|
| Facebook_348 | .669 | .669 | .730 | .847 | .426 | **.405** |
| Facebook_414 | .721 | .721 | .699 | .870 | .449 | **.420** |
| Facebook_686 | .637 | .637 | .681 | .772 | .446 | **.391** |
| Facebook_698 | .661 | .661 | .640 | .729 | **.392** | .420 |
| Facebook_1684 | .807 | .807 | .767 | .844 | .491 | **.465** |
| Facebook_3980 | .708 | .708 | .541 | .667 | .538 | **.402** |

**Table 2:** BER of feature pattern discovery comparisons on synthetic datasets: $\text{UM}^2\text{L}_{\text{ADS}}$ vs. others

| BER↓ | KM | SP | SCA | EGO | UM²L |
|---|---|---|---|---|---|
| syn1 | .382 | .382 | .392 | .467 | **.355** |
| syn2 | .564 | .564 | .399 | .428 | **.323** |
| ad | .670 | .670 | .400 | .583 | **.381** |
| ccd | .244 | .244 | .250 | .225 | **.071** |
| my_movie | .370 | .370 | .249 | .347 | **.155** |
| reuters | .704 | .704 | .400 | .609 | **.398** |

Similarly, we test feature pattern discovering ability of $\text{UM}^2\text{L}_{\text{ADS}}$ on 4 transformed multi-view datasets. For each dataset, we first extract principal components of each view, and construct sub-linkage candidates between instances with random thresholds on each single view. Thus, these candidates are various among different views. After that, the overall linkage is further generated from these candidates using "or" operation. With features on each view and the overall linkage, the goal of feature pattern discovering is to reveal responsible features for each sub-linkage. Zero-value rows/columns of learned metrics indicate irrelevant features in the corresponding group. Syn1 and syn2 are purely synthetic datasets with features sampled from Uniform, Beta, Binomial, Gamma and Normal distributions using different parameters. BER results are listed in Table 2 and $\text{UM}^2\text{L}_{\text{ADS}}$ achieves the best on all datasets. These assessments indicate $\text{UM}^2\text{L}_{\text{ADS}}$ can figure out reasonable linkages or patterns hidden behind observations, and even better than domain specific methods.

## 4.2 Comparisons on Classification Performance

To test classification generalization performance, our framework is compared with 8 state-of-the-art metric learning methods on 10 benchmark datasets and 8 large scale datasets (results of 8 large scale data are in the supplementary material). In detail, global DML methods: ITML [5], LMNN [20] and EIG [21]; local and instance specific DML methods: PLML [18], SCML (local version) [15]; MMLMNN [20], ISD [22] and SCA [4].

In $\text{UM}^2\text{L}$, distance values from different metrics are comparable. Therefore in the test phase, we first compute 3 nearest neighbors for testing instance $\tilde{\mathbf{x}}$ using each base metric $M_k$. Then $3 \times K$ distance values are collected adaptively and the smallest 3 ones (3 instances with the highest similarity scores) form neighbor candidates. Majority voting over them is used for prediction.

Evaluations on classification are repeated for 30 times. In each trial, 70% of instances are used for training, and the remaining part is for test. Cross-validation is employed for parameters tuning. Generalization errors (mean±std.) based on 3NN are listed in Table 3 where Euclidean distance results (EUCLID) are also listed as a baseline. Considering the abilities of multi-semantic description of ADS and the rigorous restrictions of RGS, $\text{UM}^2\text{L}_{\text{ADS/RGS}}$ are implemented in this comparison. Number of metrics $K$ is configured as the number of classes. Table 3 clearly shows that $\text{UM}^2\text{L}_{\text{ADS/RGS}}$ perform well on most datasets. Especially, $\text{UM}^2\text{L}_{\text{RGS}}$ achieves best on more datasets according to $t$-tests and this can be attributed to the rigorous restrictions of RGS.

**Table 3:** Comparisons of classification performance (test errors, mean $\pm$ std.) based on 3NN. $\text{UM}^2\text{L}_{\text{ADS}}$ and $\text{UM}^2\text{L}_{\text{RGS}}$ are compared. The best performance on each dataset is in bold. Last two rows list the Win/Tie/Lose counts of $\text{UM}^2\text{L}_{\text{ADS/RGS}}$ against other methods on all datasets with $t$-test at significance level 95%.

| | $\text{UM}^2\text{L}_{\text{ADS}}$ | $\text{UM}^2\text{L}_{\text{RGS}}$ | PLML | SCML | MMLMNN | ISD | SCA | ITML | LMNN | EIG | EUCLID |
|---|---|---|---|---|---|---|---|---|---|---|---|
| Autompg | **.201±.034** | .225±.031 | .265±.048 | .253±.026 | .256±.032 | .288±.033 | .286±.037 | .292±.032 | .259±.037 | .266±.031 | .260±.036 |
| Clean1 | **.070±.018** | .086±.020 | .098±.027 | .100±.027 | .097±.022 | .143±.023 | .306±.072 | .141±.024 | .084±.021 | .127±.021 | .139±.023 |
| German | .281±.019 | .284±.030 | **.280±.016** | .302±.021 | .289±.019 | .297±.017 | .292±.023 | .288±.021 | .292±.021 | .284±.014 | .296±.021 |
| Glass | .312±.043 | **.293±.047** | .389±.050 | .328±.054 | .296±.047 | .334±.050 | .529±.053 | .311±.038 | .315±.049 | .314±.050 | .307±.042 |
| Hayes-r | **.276±.044** | .307±.068 | .436±.201 | .296±.053 | .282±.062 | .378±.093 | .379±.068 | .342±.080 | .314±.072 | .289±.067 | .398±.046 |
| Heart-s | .190±.035 | .194±.063 | .365±.127 | .205±.040 | .191±.037 | .192±.036 | .203±.039 | **.186±.032** | .200±.026 | .189±.034 | .190±.030 |
| House-v | .051±.015 | **.048±.013** | .121±.240 | .066±.019 | .055±.017 | .072±.024 | .174±.075 | .063±.023 | .061±.017 | .080±.024 | .083±.025 |
| Liver-d | .363±.045 | **.342±.047** | .361±.055 | .371±.042 | .372±.045 | .364±.042 | .408±.011 | .377±.052 | .373±.045 | .380±.037 | .384±.040 |
| Segment | **.023±.038** | .029±.034 | .041±.031 | .041±.008 | .036±.006 | .063±.009 | .324±.043 | .050±.012 | .039±.006 | .059±.016 | .050±.007 |
| Sonar | .136±.032 | **.132±.036** | .171±.048 | .193±.045 | .157±.038 | .182±.038 | .220±.040 | .174±.039 | .145±.032 | .159±.042 | .168±.036 |
| W / T / L | $\text{UM}^2\text{L}_{\text{ADS}}$ vs. others | | 6 / 4 / 0 | 7 / 3 / 0 | 4 / 6 / 0 | 7 / 3 / 0 | 8 / 2 / 0 | 6 / 4 / 0 | 5 / 5 / 0 | 6 / 4 / 0 | 8 / 2 / 0 |
| W / T / L | $\text{UM}^2\text{L}_{\text{RGS}}$ vs. others | | 6 / 4 / 0 | 8 / 2 / 0 | 5 / 5 / 0 | 9 / 1 / 0 | 8 / 2 / 0 | 8 / 2 / 0 | 8 / 2 / 0 | 7 / 3 / 0 | 8 / 2 / 0 |

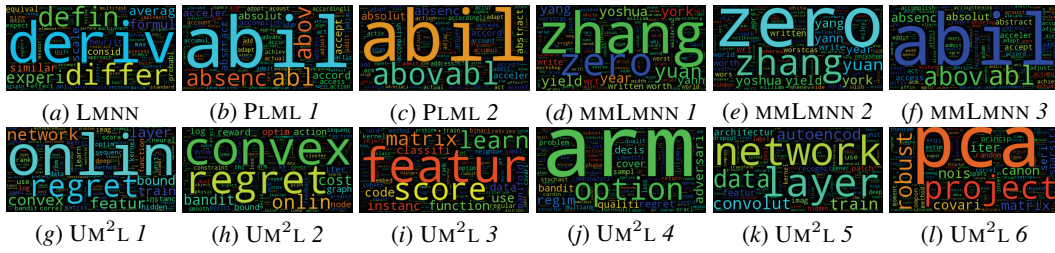

*(a)* LMNN    *(b)* PLML *1*    *(c)* PLML *2*    *(d)* MMLMNN *1*    *(e)* MMLMNN *2*    *(f)* MMLMNN *3*

*(g)* $\text{UM}^2\text{L}$ *1*    *(h)* $\text{UM}^2\text{L}$ *2*    *(i)* $\text{UM}^2\text{L}$ *3*    *(j)* $\text{UM}^2\text{L}$ *4*    *(k)* $\text{UM}^2\text{L}$ *5*    *(l)* $\text{UM}^2\text{L}$ *6*

**Figure 1:** Word clouds generated from the results of compared DML methods. The size of word depends on the importance weight of each word (feature). The weight is calculated by decomposing each metric $M_k = L_k L_k^\top$, and calculate the $\ell_2$-norm of each row in $L_k$, where each row corresponds to a specific word. Each subplot gives a word cloud for a base metric learned from DML approaches.

## 4.3 Comparisons of Latent Semantic Discovering

$\text{UM}^2\text{L}$ is proposed for DML with both localities and semantic linkages considered. Hence, to investigate the ability of latent semantics discovering, two assessments in real applications are performed, i.e., Academic Paper Linkages Explanation (APLE) and Image Weak Label Discovering (IWLD). In APLE, data are collected from 2012-2015 ICML papers, which can be connected with each other by more than one topic, yet only the session ID is captured to form explicit linkages. 3 main directions of sessions are picked up in this assessment, i.e., "feature learning", "online learning" and "deep learning". No sub-fields and additional labels/topics are provided. Simplest TF-IDF is used to extract features, which forms a corpus of 220 papers and 1622 words in total. Aiming at finding the hidden linkages together with their causes, both $\text{UM}^2\text{L}_{\text{ADS}}$ and $\text{UM}^2\text{L}_{\text{OVS}}$ are invoked. To avoid trivial solutions, regularizer for each metric is configured as $\Omega_k(M_k) = \|M_k - I\|_F^2$ for $\text{UM}^2\text{L}_{\text{OVS}}$. All feature (word) weights and correlations can be provided by learned metrics, i.e., with decomposition $M_k = L_k L_k^\top$, the $\ell_2$-norm value of each row in $L_k$ can be regarded as the weight for each feature (word). The importance of feature (word) weights is demonstrated in word clouds in Fig. 1, where the size of fonts reflects the weights of each word. Due to the page limits, supplementary materials represent full evaluations.

Fig. 1 shows the results of LMNN [20] *(a)*, PLML [18] *(b, c)*, MMLMNN [20] *(d, e, f)* and $\text{UM}^2\text{L}_{\text{OVS}}$ *(g ∼ l)* with $K = 6$, respectively. Global method LMNN returns one subplot. The metric learned by LMNN perhaps has discriminative ability but the weights of words cannot distinguish subfields in 3 selective domains. For multi-metric learning approaches PLML and MMLMNN, though they can provide more than one base metric and consequently have multiple word clouds, the words presented in subplots are not with legible physical semantic meanings. Especially, PLML outputs multiple metrics which are similar to each other (tends to global learner's behavior) and only focus on first part of the alphabet, while MMLMNN by default only learns multiple metrics with the number of base metrics equaling to the number of classes. However, results of $\text{UM}^2\text{L}_{\text{OVS}}$ clearly demonstrate all 3 fields. On session "online learning", it can discover different sub-fields such as "online convex opti-

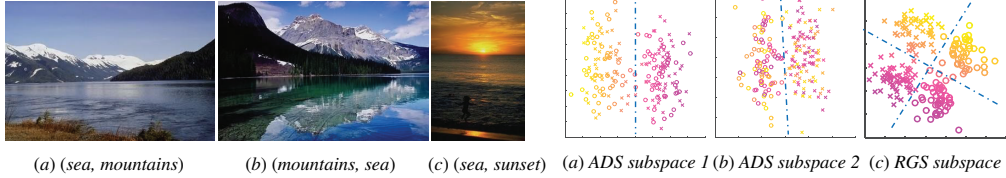

| (a) (sea, mountains) | (b) (mountains, sea) | (c) (sea, sunset) | (a) ADS subspace 1 (b) ADS subspace 2 (c) RGS subspace |

**Figure 2:** Results of visual semantic discovery on images. The first annotation in the bracket is the provided weak label. The second one is one of the latent semantic labels discovered by $\mathrm{U_M^2L}$.

**Figure 3:** Subspaces discovered by $\mathrm{U_M^2L_{ADS}}$ (a,b) and $\mathrm{U_M^2L_{RGS}}$ (c). Instances possess 2 semantic properties, i.e., color and shape. Blue dot-lines give the decision boundary.

mization" (*g* and *h*), and "online (multi-) armed bandit problem" (*j*); for session "feature learning", it has "feature score" (*i*) and "PCA projection" (*l*); and for "deep learning", the word cloud returns popular words like "network layer", "autoencoder" and "layer"(*k*).

Besides APLE, the second application is about weak label discovering in images from [23], where the most obvious label for each image is used for triplets constraints generation. $\mathrm{U_M^2L_{OVS}}$ can obtain multiple metrics, each of which is with a certain visual semantic. By computing similarities based on different metrics, latent semantics can be discovered, i.e., if we assume images connected with high similarities share the same label, missing labels can be completed as in Fig. 2. More weak label results can be found in the supplementary material.

### 4.4 Investigations of Latent Multi-View Detection

Another direct application of $\mathrm{U_M^2L}$ is hidden multi-view detection, where data can be described by multiple views from different channels yet feature partitions are not clearly provided [16]. Data with multi-view goes consistent with the assumption of ADS or RGS configuration. ADS emphasizes the existence of relevant views and aims at decomposing helpful aspects or views; while RGS requires full accordance among views. Trace norm regularizes the approach in this part to get low dimensional projection. $\mathrm{U_M^2L}$ framework facilitates the understanding of data by decomposing each base metric to low dimensional subspace, i.e., for each base metric $M_k$, 2 eigen-vectors $L_k \in \mathcal{R}^{d \times 2}$ corresponding to the largest 2 eigen-values are picked as orthogonal bases.

The hidden multi-view data [1] are composed of 200 instances and each instance has two hidden views, namely color and shape. We perform $\mathrm{U_M^2L_{ADS/RGS}}$ on this dataset with $K = 2$. Results of other methods such as SCA can be found in the supplementary material. Fig. 3 (*a*) (*b*) give the 2-D visualization results by plotting the projected instances in subspaces corresponding to metric $M_1$ and $M_2$ of $\mathrm{U_M^2L_{ADS}}$. It clearly shows that $M_1$ captures the semantic view of color, and $M_2$ reflects the meaning of shape. While for $\mathrm{U_M^2L_{RGS}}$, the visualization result of one of the obtained metrics is showed in Fig. 3 (*c*). It can be clearly found that both $\mathrm{U_M^2L_{ADS}}$ and $\mathrm{U_M^2L_{RGS}}$ can capture the two different semantic views hidden in data. Moreover, since $\mathrm{U_M^2L_{RGS}}$ requires more accordance, it can capture these physical meanings with a single metric.

## 5 Conclusion

In this paper, we propose the Unified Multi-Metric Learning ($\mathrm{U_M^2L}$) framework which can exploit side information from multiple aspects such as locality and semantics linkage constraints. It is notable that both types of constraints can be absorbed in the multi-metric loss functions with a type of flexible function operator $\kappa$ in $\mathrm{U_M^2L}$. By implementing $\kappa$ in different forms, $\mathrm{U_M^2L}$ can be used for local metric learning in classification, latent semantic linkage discovering, etc., or degrade to state-of-the-art DML approaches. The regularizer in $\mathrm{U_M^2L}$ is flexible for different purposes. $\mathrm{U_M^2L}$ can be solved by various optimization techniques such as proximal gradient and accelerated stochastic approaches, and theoretical guarantee on the convergence is proved. Experiments show the superiority of $\mathrm{U_M^2L}$ in classification performance and hidden semantics discovery. Automatic determination of the number of base metrics is an interesting future work.

**Acknowledgements** This research was supported by NSFC (61273301, 61333014), Collaborative Innovation Center of Novel Software Technology and Industrialization, and Tencent Fund.

## Footnotes

[1]Detailed derivation and efficiency comparison are in the supplementary material.

[2]This condition generally holds according to the norm regularizer in the objective function.

[3]Detailed proof can be found in the supplementary material.

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
