[Supplementary Material · supplementary.pdf]

# Supplementary Material of What Makes Objects Similar: A Unified Multi-Metric Learning Approach

**Han-Jia Ye    De-Chuan Zhan    Xue-Min Si    Yuan Jiang    Zhi-Hua Zhou**
National Key Laboratory for Novel Software Technology, Nanjing University
Nanjing, 210023, China
{yehj,zhandc,sixm,jiangy,zhouzh}@lamda.nju.edu.cn

In this supplementary material, we first prove Lemma 1 and Theorem 1 in this paper, then some implementation details are discussed. At last, additional experimental results are also presented in this material.

## 1  Reweighted Method on Symmetric $\ell_{2,1}$-norm

**Lemma 1** *The proximal problem $M'_k = \arg\min_{M \in \mathcal{S}_d} \frac{1}{2}\|M - V_k\|_F^2 + \lambda\|M\|_{2,1}$ can be solved by updating diagonal matrixes $D_1$ and $D_2$ and symmetric matrix $M$ alternatively:*

$$\{D_{1,ii} = \frac{1}{2\|\boldsymbol{m}_i\|_2}, \ D_{2,ii} = \frac{1}{2\|\boldsymbol{m}^i\|_2}\}_{i=1}^d \ ; vec(M) = (I \otimes (I + \frac{\lambda}{2}D_1) + (\frac{\lambda}{2}D_2 \otimes I))^{-1}vec(V_k) \ ,$$

*where $vec(\cdot)$ is the vector form of a matrix and $\otimes$ means the Kronecker product. Due to the diagonal property of each term, the update of $M$ can be further simplified.*

**Proof sketch**: When the row and column sparsity of a metric is needed, the regularizer of each metric can be set as $\ell_{2,1}$-norm, i.e., $\Omega_k(M_k) = \|M_k\|_{2,1}$. In both alternative and stochastic optimization processes, solving $\ell_{2,1}$-norm under symmetric constraint can be transformed to the following proximal sub-problem:

$$M = \arg\min_{M \in \mathcal{S}_d} \frac{1}{2}\|M - V\|_F^2 + \lambda\|M\|_{2,1} \ . \tag{1}$$

For simplicity, $M$ is used to denote a certain metric in $\mathcal{M}_K$, and $V$ is the intermediate solution after the gradient descent. The PSD property of a metric can be preserved by a projection after each iteration or a last-step projection [3]. Thus, only symmetric constraint is considered here, i.e., $M \in \mathcal{S}_d$. Since $\ell_{2,1}$-norm is the sum of $\ell_2$-norm on each row of $M$, which violates the symmetric requirement of a metric, directly optimizing may be time-consuming in some cases [7]. Taking symmetry into consideration, we reformulate the problem in Eq.1 as follows:

$$M = \arg\min_{M \in \mathcal{S}_d} \frac{1}{2}\|M - V\|_F^2 + \frac{\lambda}{2}\|M\|_{2,1} + \frac{\lambda}{2}\|M^\top\|_{2,1} \ . \tag{2}$$

The impact of $\ell_{2,1}$-norm is shared on $M$ and $M^\top$ equally. Taking the derivative of Eq. 2 w.r.t. $M$ and set it to zero, we can get:

$$M - V + \frac{\lambda}{2}D_1 M + \frac{\lambda}{2}MD_2 = 0 \ . \tag{3}$$

Both $D_1$ and $D_2$ are diagonal matrixes of size $d \times d$, and the $(i,i)$-th elements in $D_1$ and $D_2$ are $a_i^1 = \frac{1}{2\|\boldsymbol{m}_i\|_2}$ and $a_i^2 = \frac{1}{2\|\boldsymbol{m}^i\|_2}$, respectively. Thus, $D_1$ and $D_2$ consider $\ell_2$-norm values from *both rows and columns* of $M$. Eq. 3 is a Sylvester equation, which has a high computational cost when solved with off-the-shelf tools. To accelerate, we consider the closed form results based on Kronecker product $\otimes$, which comes to the result of Lemma 1:

$$vec(M) = (I \otimes (I + \frac{\lambda}{2}D_1) + (\frac{\lambda}{2}D_2^\top \otimes I))^{-1}vec(V) \ . \tag{4}$$

$vec(\cdot)$ means the vectorization of a matrix. It is notable that all terms in the inverse operation are in *diagonal* forms, namely $D_1$, $D_2$ and identity matrix $I$. So the inverse is not on a matrix but on a *scalar*. The $(i, j)$-element in $M$ can be calculated as:

$$M_{ij} = V_{ij}/((1 + \frac{\lambda}{2}a_j^1) + \frac{\lambda}{2}a_i^2) . \tag{5}$$

Since metric $M$ can keep symmetry in each iteration, this update flow is *symmetric projection free*. Thus, $a_i^1 = a_i^2$, which further simplifies the computation. Since $D_1$ and $D_2$ in the closed form update of $M$ in Eq. 4 also depend on the current solution of $M$, both $M$ and $D_1/D_2$ should be updated alternatively. The convergence of this reweighted method can be easily proved [8] and is validated in our additional experiments in section 4.1. In our implementation, it usually converges in about 5~10 iterations.

## 2  Proof of Theorem 1

**Theorem 1** *Suppose in the* UM$^2$L *framework, the loss $\ell(\cdot)$ is a convex one and the selection operator $\kappa_v$ in a piecewise linear form. If each training instance $\|\mathbf{x}\|_2 \leq 1$, the sub-gradient set of $\Omega_k(\cdot)$ is bounded by $R$, i.e., $\|\partial\Omega_k(M_k)\|_F^2 \leq R^2$ and sub-gradient of loss $\ell(\cdot)$ is bounded by $C$. When for each base metric[1] $\|M_k - M_k^*\|_F \leq D$, it holds that:*

$$\sum_{s=1}^{S} \mathcal{L}_{\mathcal{M}_K^s}^s - \mathcal{L}_{\mathcal{M}_K^*}^s \leq 2GD + B\sqrt{S}$$

*with $G^2 = \max(C^2, R^2)$ and $B = (\frac{D^2}{2} + 8G^2)$. Given hinge loss, $C^2 = 16$.*

**Proof sketch**: First we review notations in the above theorem. Then we figure out the relationship between two successive updates followed by the proof on the bound of sub-gradient updates.

Suppose there are totally $S$ iterations, in $s$-th iteration, the optimization problem becomes:

$$\mathcal{L}_{\mathcal{M}_k}^s = \ell_{\mathcal{M}_K^s}^s + \lambda\sum_{k=1}^{K}\Omega_k(M_k^s) = \ell(f^1(\mathbf{x}^s, \mathbf{y}^s) - f^2(\mathbf{x}^s, \mathbf{z}^s)) + \lambda\sum_{k=1}^{K}\Omega_k(M_k^s), \tag{6}$$

which is a loss function with one triplet plus regularizer for each metric. Without loss of generality, we assume in the $s$-th iteration, triplet $s$ is sampled. $\mathcal{M}_K^s = \{M_1^s, \ldots, M_K^s\}$ is the solution of current iteration, and $\mathcal{M}_K^* = \{M_1^*, \ldots, M_K^*\}$ is the optimal solution over all iterations. After a gradient or a sub-gradient descent on the loss function, the $k$-th metric is updated as:

$$M_k^{s+\frac{1}{2}} = M_k^s - \gamma^s\nabla_{1,k}^s . \tag{7}$$

$\gamma^s$ is current step-size and $\nabla_{1,k}^s$ is the sub-gradient of loss w.r.t. $M_k$. For example, when smooth hinge loss is used, as Eq. 3 *in the main text*, $\nabla_{1,k}^s = \nabla_{M_k}^s(a^s)$. Then a proximal sub-problem is solved to get a further update [4]:

$$M_k^{s+1} = \underset{M \in \mathcal{S}_d}{\arg\min} \frac{1}{2}\|M - M_k^{s+\frac{1}{2}}\|_F^2 + \gamma^s\lambda\Omega(M). \tag{8}$$

Since $M_k^{s+1}$ is the minimization of proximal subproblem in Eq. 8, we have:[2]

$$0 \in M_k^{s+1} - (M_k^s - \gamma^s\nabla_{1,k}^s) + \gamma^s\nabla_{2,k}^{s+1},$$

where $\nabla_{2,k}^{s+1} \in \frac{\partial\lambda\Omega(M_k^{s+1})}{\partial M_k^{s+1}}$ is one of the sub-gradient of the regularizer. Thus the iteration equality relationship between two successive solutions is:

$$M_k^{s+1} = M_k^s - \gamma^s\nabla_{1,k}^s - \gamma^s\nabla_{2,k}^{s+1}.$$

Since $\ell^s_{\mathcal{M}^s_K}$ is convex for each metric $M^s_1, \ldots, M^s_K$, together with the convexity of $\Omega_k(M^s_k)$, we can get [6]:

$$\ell^s_{\mathcal{M}^s_K} - \ell^s_{\mathcal{M}^*_K} + \lambda \sum_{k=1}^{K} \left[ \Omega_k(M^{s+1}_k) - \Omega_k(M^*_k) \right]$$

$$\leq \sum_{k=1}^{K} \left\langle \frac{\partial \ell^s_{\mathcal{M}^s_K}}{\partial M^s_k}, M^s_k - M^*_k \right\rangle + \left\langle \frac{\partial \Omega_k(M^{s+1}_k)}{\partial M^{s+1}_k}, M^{s+1}_k - M^*_k \right\rangle$$

$$= \sum_{k=1}^{K} \left\langle \nabla^s_{1,k}, M^s_k - M^*_k \right\rangle + \left\langle \nabla^{s+1}_{2,k}, M^{s+1}_k - M^*_k \right\rangle. \tag{9}$$

To bound the r.h.s. of above inequality, it is noticed that:

$$\|M^{s+1}_k - M^s_k\|^2_F = \|M^s_k - \gamma^s \nabla^s_{1,k} - \gamma^s \nabla^{s+1}_{2,k} - M^*_k\|^2_F$$

$$\leq \|M^s_k - M^*_k\|^2_F - 2\gamma^s \left\langle M^s_k - M^*_k, \nabla^s_{1,k} \right\rangle - 2\gamma^s \left\langle M^{s+1}_k - M^*_k, \nabla^{s+1}_{2,k} \right\rangle$$

$$+ (\gamma^s)^2 \|\nabla^s_{1,k} + \nabla^{s+1}_{2,k}\|^2_F - 2\gamma^s \left\langle M^s_k - M^{s+1}_k, \nabla^{s+1}_{2,k} \right\rangle. \tag{10}$$

Now the goal transforms to bound the last two terms of Eq. 10. Given condition $\|\nabla^s_{1,k}\|^2 \leq C^2$ and $\|\nabla^s_{2,k}\|^2 \leq R^2$, we can bound both of above two terms with $4(\gamma^s)^2 G^2$ and $G^2 = \max(C^2, R^2)$. When hinge loss is used, the loss function in the $s$-th iteration becomes:

$$\ell^s_{\mathcal{M}^s_K} = \left[ 1 + \mathrm{Dis}^2_{M^s_{k_{1,*}}}(\mathbf{x}^s, \mathbf{y}^s) - \mathrm{Dis}^2_{M^s_{k_{2,*}}}(\mathbf{x}^s, \mathbf{z}^s) \right]_+,$$

which is linear on $M_k$. Due to the constraint of each instance, i.e., $\|\mathbf{x}\|_2 \leq 1$, we have $\|\nabla^s_{1,k}\|^2_F \leq C^2 = 16$ in this case.

Plug the above relationship into Eq. 9, and sum up the loss difference for $s = 1, \ldots, S$:

$$\sum_{s=1}^{S} \mathcal{L}^s_{\mathcal{M}^s_K} - \sum_{s=1}^{S} \mathcal{L}^s_{\mathcal{M}^*_K}$$

$$= \sum_{s=1}^{S} \ell^s_{\mathcal{M}^s_K} - \sum_{s=1}^{S} \ell^s_{\mathcal{M}^*_K} + \lambda \sum_{s=1}^{S} \sum_{k=1}^{K} \left[ \Omega_k(M^{s+1}_k) - \Omega_k(M^*_k) \right] - \sum_{k=1}^{K} \left[ \Omega_k(M^{S+1}_k) - \Omega_k(M^1_k) \right]$$

$$\leq GD + 4G^2 \sum_{s=1}^{S} \gamma^s + \sum_{s=1}^{S} \frac{1}{2\gamma^s} (\|M^s_k - M^*_k\|^2_F - \|M^{s+1}_k - M^*_k\|^2_F).$$

By setting step size $\gamma^s = \frac{1}{\sqrt{s}}$ and using telescope sum [4], we can get the final result as in the Theorem 1.

## 3  Implementation Details

$\mathrm{U}\mathrm{M}^2\mathrm{L}$ is optimized in an alternative style, i.e., fixing metric set $\mathcal{M}_K$ when optimizing affiliation portion, and vice versa. We initialize the metric affiliation portion for each instance first. To determine the affiliation portion is to find which metric in $\mathcal{M}_K$ to use for a pair of instances in a particular triplet. GMM or KMeans is conducted on data with the component number the same as $K$ set in $\mathrm{U}\mathrm{M}^2\mathrm{L}$. Thus, each instance can be represented as a vector of length $K$. Elements in this vector can be regarded as a type of affiliation score of each component. Comparison between these coding values indicates which component illustrating the similar/dissimilar relationship. For instance, in Apical Dominating Similarity (ADS) with $\kappa_1 = \kappa_2 = \max(\cdot)$, given similar pair $(\mathbf{x}, \mathbf{y})$ with coding vector $(\bar{\mathbf{x}}, \bar{\mathbf{y}})$, the initial metric of this pair is selected by:

$$k = \arg\max_k \min(\bar{\mathbf{x}}, \bar{\mathbf{y}}).$$

The selection can be interpreted as a two-stage process. First, a similarity value between instances based on $K$ components is calculated in a Histogram Intersection Kernel (HIK) style [9], where

instances are similar based on some metric only when *both* of their affiliations are not small w.r.t. a certain component. Second, a metric is selected based on the largest pair-similarity value.

For stochastic optimization, according to [4], we have a general theorem as following for further accelerating the stochastic solution by deferring proximal steps.

**Theorem 2** *If $\mathbf{m}_{\hat{S}}$ is the solution after solving a succession of $\hat{S}$ self-similar proximal optimization problems for $s = 1, \ldots, \hat{S}$ as following:*

$$\mathbf{m}_s = \arg\min_{\mathbf{m}} \frac{1}{2}\|\mathbf{m} - \mathbf{m}_{s-1}\| + \lambda_s\|\mathbf{m}\|_q,$$

*and $\mathbf{m}^*$ is the optimal solution for the accumulated proximal problem:*

$$\mathbf{m}^* = \arg\min_{\mathbf{m}} \frac{1}{2}\|\mathbf{m} - \mathbf{m}_0\| + \sum_{s=1}^{\hat{S}} \lambda_s\|\mathbf{m}\|_q.$$

*Then for $q \in \{1, 2, \infty\}$, $\mathbf{m}_{\hat{S}}$ and $\mathbf{m}^*$ should be identical.*

With this theorem, for some frequently used norms we can use a *lazy update* rule which only performs proximal operator after a number of sub-gradient descent operations rather than solving the proximal sub-problem once the sub-gradient operation is finished.

Four types of similarity with $\text{UM}^2\text{L}$ are discussed, namely Apical Dominating Similarity (ADS), One Vote Similarity (OVS), Rank Grouping Similarity (RGS) and Average Case Similarity (ACS), which have different optimization properties. The RGS and ACS are convex and thus alternative approach can get global optimal solutions. The OVS is a non-convex one, whose results depend on initializations. The ADS is a semi-convex problem [5], i.e., the objective of ADS is convex when the affiliations of similar pairs in triplets are fixed. To better utilize the semi-convex property, we compute only the affiliations of similar pairs in the initialization phase and following an alternative process. Since when these affiliations are fixed, the whole problem becomes a convex one even there still exists a $\max(\cdot)$ operator. Therefore, both the metric set $\mathcal{M}_K$ and the affiliations of dissimilar pairs can be jointly optimized.

## 4 Additional Experimental Results

To show different properties of $\text{UM}^2\text{L}$, some more experimental settings and results are described in this section. Our experiments are performed on a cluster of 32 machines, each of which has four 6-core 2.53GHz CPUs and 48G RAM.

### 4.1 Convergence of Reweighted Method on Symmetric $\ell_{2,1}$-norm

Since $\ell_{2,1}$-norm is often used to force row/column sparsity on learned metrics, we first validate the convergence property of the reweighted method proposed in Section 1. Given different sizes of input metrics, we test the change of the objective function in Eq. 2. Two sizes of metrics are showed in Fig. 1: the first is a $20 \times 20$ metric (*a*) and the second is of size 200 (*b*).

It can be found that: First, the reweighted method converges at last; Second, this method converges very quickly and the time to converge may depend on the size of the input. The larger the metric the more iterations it needs to converge. When the dimension is low, it can converge in about 5 iterations. While with a larger input of size 200, the objective value converges in about 10 iterations.

In addition, on the larger size input, time comparison is conducted between the reweighted method and the symmetric iterative projection one in [7]. We force the two methods to achieve nearly the same objective value in Eq. 1 at last and the time comparison is conducted 100 times in total. The average convergence time (in second) for reweighted one and symmetric projection one are 0.0093 (0.0012) and 0.0665 (0.0023) respectively. Values in brackets are standard deviations of the 100-time values. This result shows our reweighted solver can be more efficient when dealing with $\ell_{2,1}$-norm regularization on metrics, since it takes norm on row and column into consideration simultaneously and is symmetric projection free.

(a) *small metric*  (b) *large metric*

Figure 1: Convergence results of the reweighted method on symmetric $\ell_{2,1}$-norm. Changes of objective values on two different sizes (dimensions) of input metrics are showed.

## 4.2 Comparison on Classification Performance

To test the classification ability of $\text{UM}^2\text{L}$, comparisons with state-of-the-art DML methods are conducted on some larger datasets. 3NN is used to test the performance of each method. Each dataset is randomly split into three parts, 40% for training, 30% for validation, and the rest is for test. Each method first tunes parameters on the validation set and retrains the model with the best parameters on the combination of training and validation data, then the model is used for test.

For $\text{UM}^2\text{L}$, Apical Dominating Similarity (ADS) and Rank Grouping Similarity (RGS) specifications are used. For the first one, it tries to explain similarity between instances with different semantics or local reasons. For the latter one, it can be regarded as rigorous constraints on each learned metric that given triplets should be satisfied over all learned metrics. Thus, it will facilitate the last prediction process with $\mathcal{M}_K$. The results are showed in Fig. 2. Four groups of methods, namely $\text{UM}^2\text{L}$, local DML, global DML and baseline $k$NN with Euclidean distance (denoted as EUCLID), are separated by spaces. For local DML methods, the number of metrics is set the same as the number of classes. If a method cannot give a result in 24 hours, we set its test error the same as the worst one in the others and denote it as "N/A".

From comparison results, $\text{UM}^2\text{L}_{\text{ADS/RGS}}$ can achieve best results on 7/8 datasets. In general, the RGS version performs better, but it may overfit sometimes, e.g., in *Reut8*. MMLMNN, a local metric learning method which trains a metric for each cluster, is also competitive on most datasets since it considers the spatial linkages between heterogeneous instances. SCML (local version) first constructs bases from data and then builds a parametric function to assign metric for each instance. The performance of SCML depends on the bases selection and optimization process. The results of SCA are not stable, since its optimization may fall into local optimal, which highly depends on the initialization. ISD and PLML consume a lot on large datasets. In particular, ISD, which learns a metric for each instance in a transductive way and has a high computational burden, intends to overfit in most cases. The global metric learning methods, i.e., ITML, EIG and LMNN can also get satisfied results. In summary, Local DML methods have the ability to cover space and semantic locality in the learning process, which leads to better performance compared with global DML methods. But they are prone to overfit when the dimensionality is low, e.g., on *Page_blocks*.

More investigations are conducted to reveal the influences of the number of metrics on classification performance. Fig. 3 gives test errors on different datasets when $\text{UM}^2\text{L}_{\text{ADS/RGS}}$ are provided with different number of metrics. From the results, we can find that there are turning points on performance curves, i.e., when the number of metrics increases, the test error decreases at first and then increases. This could be attributed to the model complexity: the model becomes powerful with the number of metrics increase, and then it is prone to overfitting. From the empirical results, it is suggested that by configuring $K$ (the number of metrics) close to the number of classes $C$ (when the number of latent semantics is known), the model can return satisfactory results in most cases.

Figure 2: Classification test error of $\text{UM}^2\text{L}$ with other $\text{DML}$ methods on 8 datasets. $\text{UM}^2\text{L}$ based on ADS and RGS are placed at first. Followings are local and global based methods. Last is Euclidean 3NN baseline. Groups are separated by spaces.

Figure 3: The change of classification performance (test error rate) when the number of metrics learned in $\text{UM}^2\text{L}_{\text{ADS}}$ and $\text{UM}^2\text{L}_{\text{RGS}}$ is changed. Bracket after the name of each dataset shows the number of classes ($C$).

(a) LMNN metric    (b) PLML metric 1    (c) PLML metric 2    (d) MMLMNN 1

(e) MMLMNN 2    (f) MMLMNN 3    (g) SCA metric 1    (h) SCA metric 2

(i) SCA metric 3    (j) SCA metric 4    (k) SCA metric 5    (l) SCA metric 6

(m) UM$^2$L$_{ADS}$ metric 1    (n) UM$^2$L$_{ADS}$ metric 2    (o) UM$^2$L$_{ADS}$ metric 3    (p) UM$^2$L$_{ADS}$ metric 4

(q) UM$^2$L$_{ADS}$ metric 5    (r) UM$^2$L$_{ADS}$ metric 6    (s) UM$^2$L$_{OVS}$ metric 1    (t) UM$^2$L$_{OVS}$ metric 2

(u) UM$^2$L$_{OVS}$ metric 3    (v) UM$^2$L$_{OVS}$ metric 4    (w) UM$^2$L$_{OVS}$ metric 5    (x) UM$^2$L$_{OVS}$ metric 6

Figure 4: Word clouds generated from the results of compared DML methods. The size of word depends on the importance weight of each word (feature). The weight is calculated by decomposing each metric $M_k = L_k L_k^\top$, and calculate the $\ell_2$-norm of each row in $L_k$, where each row corresponds to a specific word. Each subplot gives a word cloud for a base metric learned from DML approaches. LMNN learns a global metric, therefore there is only one subplot for it. Metric bases learned by PLML is similar to each other, so we only present two of them. MMLMNN by default learns 3 metrics, which is the same as the number of classes. SCA and UM$^2$L both learn 6 metrics.

## 4.3 Comparisons of Latent Semantic Discovering

To investigate the ability of latent semantics discovering for UM$^2$L, two assessments in real applications are performed, i.e., Academic Paper Linkages Explanation (APLE) and Image Weak Label Discovering (IWLD).

For APLE, we test UM$^2$L$_{ADS/OVS}$ on a four-year ICML paper dataset. We use real-valued TF-IDF features and each feature corresponds to a word in the paper corpus. The metric learned by DML reveals a type of weights on different features. Given a learned metric $M_k$, we first do a decomposition $M_k = L_k L_k^\top$, and the $\ell_2$-norm value of each row of $L$ can be regarded as the weight for each feature (word). Thus, local metric learning methods can get multiple metrics (weights of words) for different semantics. Results of each method are demonstrated in word clouds in Fig. 4, where the size of a word is in proportion to its weights.

Besides the results of global metric learning methods LMNN, local methods PLML, MMLMNN and our proposed UM$^2$L$_{OVS}$, the results of SCA and UM$^2$L$_{ADS}$ are added. The number of metrics learned by SCA is also set to 6. SCA can discover some key words in "online learning" and "deep

| (a) (sea, mountain) | (b) (mountain, sea) | (c) (plant, sunset) | (d) (sunset, plant) |
| (e) (plant, mountain) | (f) (plant, mountain) | (g) (sunset, mountain) | (h) (desert, sunset) |
| (i) (plant, sunset) | (j) (mountain, plant) | (k) (desert, mountain) | (l) (desert, plant) |

Figure 5: Results of visual semantic discovery on images. The first annotation in the bracket is the provided weak label, and the second one is one of the latent semantic labels discovered by $\text{UM}^2\text{L}$.

learning" fields, such as "reward", "bound" and "adversary" about online learning as well as "GPU", "layer" for deep learning. For $\text{UM}^2\text{L}_{\text{ADS}}$, 3 main session semantics are also discovered, but weights of words are different from $\text{UM}^2\text{L}_{\text{OVS}}$. From subplot (m), the main topic should be domain adaptation, which can be related to transferrable feature learning in deep learning researches. Subplots (n), (o) and (q) are all about feature learning but with different subfields, i.e., (n) should be mainly about structure feature learning in DML, and pairwise constraints between items are emphasized; (o) is about manifold learning in feature construction and (q) is about subspace learning and dimensionality reduction in feature learning, with the key word "eigenvector" being emphasized. Subplot (p) is related with deep learning, the word cloud clearly shows the item "network layer", "RBM", etc. The last subplot (r) is definitely about online learning, with key words "arm", "optimal", "bandit" and "regret". It is notable that latent semantics are discovered from leaned metric, which validate the discriminative ability of $\text{UM}^2\text{L}$ over semantics. It will benefit some subsequent tasks such as classification.

Besides APLE, $\text{UM}^2\text{L}$ can also be applied to Image Weak Label Discovering (IWLD). For an image, there may be multiple complex semantics. The linkage between two images may only depend on one of their shared semantic, while the disconnection between two images may also lie in a particular semantic. We use an image dataset from [10] and the original dataset contains five labels, namely: desert, mountain, sea, sunset, and plant. For each image, we select its most obvious label and transform this dataset such that each instance only has a single label. Thus, in this case, although images are with plenty of latent semantics, similarities between images are just based on one of them. By computing similarities based on different learned metrics, latent semantics can be discovered, i.e., if we assume images connected with high similarities share the same label, missing labels can be completed.

In our experiments, $\text{UM}^2\text{L}_{\text{OVS}}$ is used with regularizer $\Omega_k(M_k) = \|M_k - I\|_F^2$ to avoid trivial solutions. Results of IWLD are showed in Fig. 5, where the image annotations in brackets indicate the pair of training label and discovered label. One image may have complex semantics. For instance, image (b) is about a lake beside a mountain and image (j) is a picture of mountains and trees. They are similar since both of them have mountains (similar since they are computed with the metric for semantic "mountain"). Image (a) is also about mountains, but the lake is more obvious. Given the training label "sea", it is hard to link with pictures with "mountains". $\text{UM}^2\text{L}_{\text{OVS}}$ can learn multiple metrics, one for each semantic. If images (a) and (b) are dissimilar in the supervision triplets,

Figure 6: Subspaces discovered by MVTE ($a,b$) given original triplets by and SCA ($c,d$) with triplets and data, respectively. Subspaces discovered by SCA, $\text{UM}^2\text{L}_{ADS}$ and $\text{UM}^2\text{L}_{RGS}$ under noisy data and original triplets are showed in ($e$) - ($j$). Instances possess 2 semantic properties, i.e., color and shape. Blue dot-lines give the decision boundary (best viewed in color).

$\text{UM}^2\text{L}_{OVS}$ finds one metric (may be about "sunset") to explain their disconnection, and does not deny their similarity over metrics about "sea" and "mountain".

## 4.4 Investigations of Latent Multi-View Detection

This subsection gives more results on the ability of latent multi-view detection for $\text{UM}^2\text{L}$. Trace regularizer is used to get low rank projections with the learned metrics. Original data come from [1] and there are two views (semantics), namely, the color view and the shape (circle and cross in the following figures) view. Triplets are generated from original data as side information and feature used are the combination of two views. The number of learned metrics is set to 2.

MVTE [1] generates 2-D representation of data only based on triplets, which gives 2-view results in subplot ($a$) and ($b$) in Fig. 6. It can be found that these two representations of data correspond to shape and color, respectively. However, there are some outliers, which reduce the discriminative ability in each view. SCA [2] is also compared. With linkages from triplets and combined original features, it can produce two low rank projections and get results in Fig. 6 ($c$) to ($d$). View1 reflects the shape semantic and View2 shows the color one. Together with the results of 2-view representations of $\text{UM}^2\text{L}_{ADS/RGS}$ in the main text of this paper, it can be found that $\text{UM}^2\text{L}$ can construct semantic low rank representations of combined data. Moreover, methods learning with side information as well as combined features such as $\text{UM}^2\text{L}$ and SCA can find better representations.

To better compare $\text{UM}^2\text{L}$ and SCA, we test them under a noisy environment. By concatenating original data with 10 dimension random noise from [0,1], both methods learn projections together with true triplet information under this difficult environment. Results of $\text{UM}^2\text{L}_{ADS}$ and SCA are showed in subplots ($e$) - ($h$) in Fig. 6. In this hard scenario, SCA can find the color view as well, but in shape view (View2, ($f$)), two shapes are not easy to determine, especially for instances near the boundary. $\text{UM}^2\text{L}_{ADS}$ still finds color and shape views, and its projections are easy for boundary determination. Thus, $\text{UM}^2\text{L}$ is robust to noise. It can extract useful features to explain the given triplet constraints and discover different semantics.

$\text{UM}^2\text{L}$ with RGS is also tested on the noisy data. Different from the ADS version which assumes different views possess different semantics to build a linkage, RGS version restricts similar/dissimilar consistency among all views. The noisy results from Fig. 6 ($i$) ($j$) are different from the result in the main paper. In Fig. 3 ($c$) *in the main text*, both color and shape linkages can be explained in a 2-D projection learned by $\text{UM}^2\text{L}_{RGS}$. In this noisy case which is hard to classify in both views, $\text{UM}^2\text{L}_{RGS}$ generates two different views. It is notable that the results of $\text{UM}^2\text{L}_{RGS}$ not only reflect two views of data (color and shape) but also have high discriminative ability, which results from the rigorous constraints of $\text{UM}^2\text{L}_{RGS}$.

## Footnotes

[1]This condition generally holds according to the norm regularizer in the objective function.

[2]For $M_k$ in $\mathcal{S}_d$, symmetric of $M_k$ can be guaranteed by the gradient of Eq. 1 with $\Omega(M)$ configured as symmetric norms, e.g., trace norm or $\frac{1}{2}(\|M\|_{2,1} + \|M^\top\|_{2,1})$, therefore 0 can be a valid sub-gradient of the objective function in proximal subproblem.