[Reviews · NeurIPS 2016]

Reviewer 1

Summary

This paper deals with distance metric learning and proposes a general multiple metric learning approach, where several distance measures are learned and combined together in a certain way (to be defined by the user) so as to reflect different localities or semantic relations in the data. The proposed formulation aggregates several previously introduced ideas into a general formulation, and the authors suggest a (stochastic) proximal gradient algorithm to solve it. Extensive numerical experiments are displayed to assess the relevance of the proposed approach in various tasks.

Qualitative Assessment

The main contribution of this paper is to introduce the interesting idea of "activator" (or integrator) operating on multiple metrics. Depending on how this activator (kappa) is picked, one may recover some previously proposed multiple metric learning approaches or create new relevant ones such as those discussed in Section 2.1. The proposed algorithm/analysis is an application of stochastic proximal (sub)gradient descent and is thus not a significant contribution from the optimization point of view (and the presentation of the algorithm is actually quite confusing, see below). My main problem with the paper is the experimental section. I was expecting some experiments showing how the choice of kappa influences the type of metrics that are learned, with an analysis of why some choices are better suited to some kind of problems / datasets / networks. Part of such an analysis could have been performed on cleverly designed synthetic networks. Instead, we get a series of confusing experiments on datasets without much description or analysis: - Section 4.1 lacks a lot of details on the experimental setup (what are the training/test sets? how are training triplets constructed? etc.). Furthermore, there is no analysis of the results: why is UM2L doing so much better than competing methods on these datasets? - I am a bit skeptical about the results of Section 4.2. The Euclidean distance outperforms LMNN and SCA on 8/10 datasets, which is very surprising to me given that these datasets are low-dimensional. Of course they are relatively small so overfitting could maybe be an issue, but then UM2L should overfit even more as it learns many more parameters than LMNN. Furthermore, this is also true for the experiments on larger-scale datasets shown in the supplementary. My guess is that the hyperparameters of competing methods have not been carefully tuned. Can the authors specify precisely which hyperparameters were tuned and in which range/grid? - The results of Section 4.3 and 4.4 are not evaluated by any quantitative measure and it is thus hard to judge whether the results are good or not. The experiment on weak image labeling does not include any competing method and is described in 5 lines, making it impossible for the reader to understand what is going on. Some other comments: - The choice of using triplet constraints is a little bit counterintuitive: as the approach is motivated by linkage problems, it would seem more natural to consider pairwise constraints (linked / not linked). One could then define a kappa function for each type of pairs, just like what is done in the current version. Any reason why the authors chose to work with triplets? - In Section 2.1, it is not obvious why the OVS model is relevant in practice. It would be nice to have a motivational example. - In Section 2.1, the authors should add a paragraph on locality-related multiple metrics, where they mention the RBF function and the use of indicator function of class/cluster. Note that for this last case, there is a slight problem of notation since kappa is supposed to operate only on the similarity scores, not on side information. - The end of Section 2.1 deals with regularization (not the choice of kappa) and should be moved elsewhere. - Section 2.2 is quite messy. The authors alternate between different regularizers, loss functions (general or hinge loss), projection strategies (at each iteration or at the end), etc. In the end it is not clear what the algorithm actually is and for which choices of the above Theorem 1 holds. I think the authors should prove their result in the most general setting they can handle (e.g.: kappa piecewise linear, convex differentiable loss, regularizer with efficient proximal operator) and then specialize it when needed. Also, Theorem 1 does not mention any condition on the step size sequence, which is definitely needed for the result to hold. - First paragraph of Section 4: the authors mention "targets" and "impostors". Unless the reader is familiar with LMNN, this is impossible to understand what is meant here. Note that choosing these implies the knowledge of instance labels (which are not available in all presented experiments), and an initial distance. Please clarify. To summarize, I think this paper introduces some interesting ideas but is not mature enough yet for publication at NIPS. Typos: - Line 68: A^t_{xy} is in \mathcal{S}_d^+ - Line 81: f_{\mathcal{M}_K} is undefined - Line 102: dissimilarty

Confidence in this Review

3-Expert (read the paper in detail, know the area, quite certain of my opinion)


Reviewer 2

Summary

This paper presents a framework for multi-metric learning that unifies some previous methods in the literature. The authors advocate a particular choice in this framework (a 2,1 matrix norm regularizer) and show both quantitative and qualitative results on real data.

Qualitative Assessment

This paper presents a "unified" framework for multi-metric learning, which is presented as equation 2. It's not clear how unifying it is. It seems to be intended to be an extension of other general Mahalanobis metric learning frameworks (see recent surveys on metric learning e.g. "Metric Learning: A Survey") but focused on the multi-metric case. It assumes triplet constraints (which is somewhat restrictive for a unified approach) and discusses various choices for regularization and combination of multiple metrics (e.g., min, max, and average). The paper feels almost more like a survey article than a NIPS paper---the framework presented is clearly incremental, straightforward, and it is not obvious what we have actually gained from this generalization. Ultimately, the authors seem to advocate the use of a 2,1 matrix-norm regularizer, and use it for their experiments. They present results of three flavors: quantitative comparisons to non-metric learning methods, quantitative comparisons to metric learning methods, and qualitative comparisons on some additional data. Interestingly, it seems that the only multi-metric learning comparison that was made was to MM-LMNN. There are many other potential local metric learning methods that could be more relevant than the single metric methods chosen (see, e.g., "A Survey on Metric Learning for Feature Vectors and Structured Data" for many examples). So at the end of the day we have a fairly minor tweak on standard metric learning formulations (use of multiple metrics + 2,1-norm regularization) plus a set of reasonably compelling experiments. Given the vast literature on the topic of metric learning, I think the bar is too high for a paper such as this to be accepted to a conference like NIPS. ====== After rebuttal: Thanks for clarifying that there were in fact comparisons to other local metric learning methods. I raised the technical quality score but kept the other scores.

Confidence in this Review

3-Expert (read the paper in detail, know the area, quite certain of my opinion)


Reviewer 3

Summary

This paper presents a framework - UM2L - for learning multiple metrics at the same time for then applying a kind of nearest neighbor classification or a majority voting. The framework is tailored to triplet-based constraints and allows to integrate different loss functions, different functions to combine the metrics that can depend be adapted to similar instances or impostors and various regularizers even though the paper considers mainly the L-2,1 norm. The problem is formalized as a general optimization problem for which a theorem of convergence is given in the context of gradient descent. Some examples illustrating the interest of the framework are given. A large experimental evaluation is presented.

Qualitative Assessment

Pros -Generality of the framework for multiple metric learning allowing to deal with different metric combination -Wide range of applicability and good empirical results -Theoretical result on the convergence of the optimization method -paper well written Cons -The framework does not really provide a solution for combining metrics learned in different regions that may suffer from discontinuity between two regions. So there is no particular novelty in the combination of local metrics. -The method requires a specific initialization that is not really discussed or analyzed -No theoretical guarantees in terms of generalization capability Overall: Interesting paper for combining multiple metrics that offer a wide range of application. The paper has no theoretical guarantees and does not really bring a solution to the problems posed by local metric learning methods. Comments -I like the proposed framework because it embeds a large set of possible way of combining metrics. The idea of using different functions for combining metrics could have been used in other papers, for me the real novelty is to provide a sufficiently general framework for combining these metrics. Even though, it must be noticed that combining multiple metrics with a max function has not really been studied, as far as I know. -The initialization of the model is evoked quickly in the supplementary material but it is not really discussed in the paper. This initialization is based on a clustering step which has a probably an important impact on the result and deserves more attention/analysis. -Some local metric learning approaches try to learn a metric for a local region to capture local regularities. However, at test time, the choice of the metric to use is a critical issue because of the discontinuity between the learned metrics in case where the two instances to compare may belong to two different regions. Indeed, even for two close regions the metric can be rather different and combining them is not really easy. This is a difficult task, but this paper does not seem to provide a solution to this problem, check this reference for example: *V. Zantedeschi, R. Emonet, M. Sebban. Metric Learning as Convex Combinations of Local Models With Generalization Guarantees. The IEEE Conference on Computer Vision and Pattern Recognition (CVPR), 2016. At test time, if I understood correctly, the paper uses all the metrics for all the pairs of examples without penalizing the metrics wrt to spatial information, which seems not natural for taking into account local similarities. I understand that each metric used in UM2L can capture different semantic information in the data, but I do not see how local information can be captured if the learning(or testing) step is global to the whole dataset. -In the experimental comparison - Section 4.2 in particular, UM2L is used with a specific setting: use of the mixed norm and a majority vote over 3*K distances. The other approaches, in particular the local ones, do not use such a setting and it would be interesting to evaluate if this really the framework that is better or the procedure to take the decision. For example, it could be interesting to evaluate the performance of each individual metric which can help to assess the expressiveness of the model. The impact of regularization can also be analyzed. -In the experimental setup, it is mentioned that 3 targets en 10 impostors are built for each dataset, but we do not know how these triplet are chosen (randomly, by a landmark selection method, cluster centers, ...) -In the multiview setting, an assumption often made is that some views may compensate other views when they are wrong. In this setting, if one can not bring different weights to each view, it not clear if the combination can bring a better information than individual view. -When considering the average similarity (ACS), in the batch setting, it is not clear for me if the framework is useful in a 1-view setting if one initializes the model to the same metric before optimization. I do not see how the model can learn different metrics unless we force the triplets to be optimized only on some particular metrics. -There is no theoretical guarantees in terms of generalization ability to new instances. I think that this is probably an important line of work to consider. In particular, nothing is said about the importance of maximizing the margin or in other words on the separation ability brought by each metric. However, this seems an important aspect in particular with RGS and OVS settings. In OVS, for example, trying to maximize the margin can be way to avoid trivial solutions. On the other hand, using a ||M_k-I|| regularization can be a good idea in problems where the Euclidean distance works well. -l.90, notation "\kappa s" has not been introduced, is it \kappa_v ?.

Confidence in this Review

3-Expert (read the paper in detail, know the area, quite certain of my opinion)


Reviewer 4

Summary

The paper considers contexts where similarity relationships between objects can be described with multiple latent semantics (see examples given in lines 29 to 36 and Section 2.1). According to the paper, existing metric learning approaches usually consider contexts where a single metric is learned for the whole training dataset, or where multiple metrics are defined locally (for instance, a specific metric is learned for each category of the dataset) but with a single semantic meaning. The authors then define a Unified Multi-Metric Learning approach which integrates the consideration of linking semantic ambiguities and localities in a single framework. In particular, their method generalizes existing approaches and can learn metrics that exploit different representations of the data (i.e. each of the metrics reflects a specific type of inherent spatial or semantic properties of objects).

Qualitative Assessment

I voted "Sub-standard for NIPS" but my opinion is actually borderline. There is no particular theoretical or algorithmic novelty in the paper, the main novelty lies in the type of tasks that account for multiple latent semantics in the data. It seems to me that the submitted paper is very similar to ref [a] (see questions below) that already proposed to tackle similar types of problems in the context of multi-modal data (e.g. multi-media data that can be represented with audio, video and text description). If this the case, then the contribution of the paper seems limited to me. Concerning experiments, the method is tested in different contexts which seem relevant. The explanations are clear and the paper is well written. My main concerns about the paper are the following: 1) The model described in Section 2 is very similar to the approach described in ref [a, Section 4]. In particular, the multiple latent semantics (e.g. multiple views in Section 4.1) can be seen as different modalities of heterogeneous data in [a] (e.g. video, text and audio in multi-media data). In [a, Section 4], the authors propose different ways of combining nonlinear Mahalanobis distance models (see [a, Table 1]) to compare heterogeneous data, and one of them corresponds to Average Case Similarity (ACS) described in Section 2.1. Can ref [a] really be seen as a way to solve the problem of multiple latent semantics? If this is the case, what is the contribution of the submitted paper compared to [a]? Otherwise, I agree with the authors that most metric learning approaches consider only one type of description for objects. 2) There is a mistake in Section 2.2. If kappa_1 = min(.) or kappa_2 = max(.) in Eq. (2) (which is the case for ADS and RGS) then the global problem in Eq. (2) is nonconvex and nondifferentiable. As mentioned in lines 148 to 153, the different matrices M_k can be fixed and the problem can be decomposed as convex upper bounds. However, contrary to what is stated in lines 154-155, there is no theoretical guarantee that the algorithm converges to a local optimum or a stationary point (see for example refs [b] or [c]) of the nonconvex problem although the objective value can be decreased at each iteration of the proposed algorithm. Theorem 1 does not prove that the algorithm converges to a local optimum or a stationary point of the nonconvex problem either, the proof only applies to the convex upper bound (i.e. when the different matrices M_k are fixed). References: [a] McFee, B. and Lanckriet, G.R.G., Learning multi-modal similarity, Journal of Machine Learning Research (JMLR) 2011 [b] Kiwiel, K. C., Methods of descent for nondifferentiable optimization. Lecture Notes in Mathematics, Springer, 1985 [c] Bertsekas, D. P., Convex Optimization Algorithms. Athena Scientific, 2015

Confidence in this Review

2-Confident (read it all; understood it all reasonably well)


Reviewer 5

Summary

In this paper, the authors introduce a framework for multi-metric learning. Different from conventional approaches to distance metric learning, which mainly focus on utilizing one single type of links, this paper proposed a unified framework for utilizing multiple types of links in distance metric learning, and provided a good solution for the new problem.

Qualitative Assessment

Strengths: * I liked the idea of learning multiple distance metrics to utilize multiple different types of similarities in the data. It seems to be a more practical setting for distance metric learning. Conventional distance metric learning usually requires a set of "ground-truth" constraints (must-link and cannot-links), which can be hard to get in many real-world applications. It is very positive that the author(s) properly discussed the properties of the UM2L framework and validated in the experiments. * A general framework for multiple metric learning is proposed. In addition to unifying some existing (multi-) metric learning methods, it can also extend to diverse types of similarities with both spatial and semantic properties considered. Convergence and acceleration of UM2L are also discussed. * Experiments from diverse domains are interesting. Besides focusing on classification results, this paper applies the proposed UM2L framework in various real applications. Suggestions: * Compared to traditional distance metric learning where the goal is to learn a single matrix M, the number of parameters that the proposed method needs to estimate is much larger. Although the authors briefly discussed about the execution time of the method in the supplementary material, it might be better if a more thorough analysis is made on the time complexity of the proposed method so it is easier to see how it compares with other methods. * Since different subsections of experiments contain different applications, it would be better to give more descriptions and explanations on real tasks and results. *Diverse properties of the defined similarities and the unified view of UM2L can be demonstrated by plots/figures.

Confidence in this Review

3-Expert (read the paper in detail, know the area, quite certain of my opinion)


Reviewer 6

Summary

This paper presents UM^2L a Unified Multi-Metric Learning framework. The idea behind this framework is to jointly learn several metrics where each metric can capture a specific spatial or semantic property of the data. In the framework the loss function and two functional operators can be chosen depending on the application. A general solution is given. The experiments demonstrate the performance of the framework on 4 different tasks. These tasks are social linkage/feature pattern discovering, classic metric learning with classification, discovery of latent semantics and discovery of latent views.

Qualitative Assessment

Summary of the comments: The paper is well-written and the proposed framework is well analysed. The experiments are extensive and demonstrate the adaptability of the approach to different applications. Technical Quality: The paper proposes an analysis of the framework and several possible instantiations, covering a large range of settings, are proposed. A general solution is given and is theoretically analysed. Furthermore the experiments demonstrate the good performance of the approach in a wide range of settings. - The loss function and the functional operators seem to be important parts of the framework (lines 94-95). The discussion on the choice on the functional operators is extensive and proposes several possibilities. However there is no such discussion on the loss function (and in the analysis it is often assumed that this loss is the smooth hinge loss). Novelty/originality: The paper proposes a novel method for multi-metric learning. One of the originalities is to parametrize the combination of the different metrics with a functional operator which can be selected with respect to the application at hand. Potential impact or usefulness: The framework looks promising as it is quite general and could be adapted to several problems. Furthermore several applications are proposed in the experiments. Clarity and presentation: The paper is well written but some aspects of the work could be clarified: - Lines 81-82 the fact that there is two functional operators is misleading and their role is hard to understand. It could have been interesting to present them as functions able to combine the multiple metrics. Similarly, applying \kappa_v to f_{\mathcal{M}_K} is misleading as it hardly reflect the fact that it corresponds to a combination of multiple metrics. - In the experiments there is no indication on the considered loss function. Is it the smooth hinge loss as in the analysis ? - I think that the discussion on the optimality of the solution (lines 79 to 86 in the supplementary) should be in the main paper as it can be a critical factor when choosing the functional operators. For example, when using UM^2L_{OVS} in the experiments, are the results reported obtained after one initialization ? or several ? - Minor remark: Lines 279 and 281 I believe that Fig. 2 is in fact Fig. 1 ?

Confidence in this Review

2-Confident (read it all; understood it all reasonably well)